evolution/environmental science

cooking pottery, hunter–gatherers, organic residue analysis, circum-Baltic area, Late Mesolithic, Early Neolithic

**Authors for correspondence:**
Blandine Courel
e-mail: bcourel@britishmuseum.org
Oliver E. Craig
e-mail: oliver.craig@york.ac.uk

[†]These authors contributed equally to this work.

# Organic residue analysis shows sub-regional patterns in the use of pottery by Northern European hunter–gatherers

Blandine Courel[1,†], Harry K. Robson[2,†],
Alexandre Lucquin[2,†], Ekaterina Dolbunova[1,3],
Ester Oras[4], Kamil Adamczak[5], Søren H. Andersen[6],
Peter Moe Astrup[6], Maxim Charniauski[7],
Agnieszka Czekaj-Zastawny[8], Igor Ezepenko[7],
Sönke Hartz[9], Jacek Kabaciński[10], Andreas Kotula[11],
Stanisław Kukawka[5], Ilze Loze[12], Andrey Mazurkevich[3],
Henny Piezonka[13], Gytis Piličiauskas[14], Søren
A. Sørensen[15], Helen M. Talbot[2], Aleh Tkachou[7],
Maryia Tkachova[7], Adam Wawrusiewicz[16],
John Meadows[17], Carl P. Heron[1] and Oliver E. Craig[2]

[1]Department of Scientific Research, The British Museum, London WC1B 3DG, UK
[2]BioArCh, Department of Archaeology, University of York, York YO10 5DD, UK
[3]The State Hermitage Museum, 34 Dvortsovaya Embankment, Saint Petersburg 190000, Russian Federation
[4]Institute of Chemistry, University of Tartu, Ravila 14A, 50411 Tartu, Estonia
[5]Institute of Archaeology, Nicolaus Copernicus University, Szosa Bydgoska 44/48, 87-100 Toruń, Poland
[6]Moesgård Museum, Moesgård Alle 15, Højbjerg 8270, Denmark
[7]Department of Archaeology of Prehistoric Society, Institute of History, National Academy of Sciences of Belarus, Academic St 1, 220072 Minsk, Belarus
[8]Institute of Archaeology and Ethnology, Polish Academy of Science, Sławkowska 17, 31-016 Krakow, Poland
[9]Stiftung Schleswig-Holsteinische Landesmuseen, Schloss Gottorf, 24837 Schleswig, Germany
[10]Institute of Archaeology and Ethnology Polish Academy of Science, 31-016 Kraków, Poland
[11]Seminar für Ur- und Frühgeschichte, Georg-August-Universität Göttingen, Nikolausberger Weg 15, 37073 Göttingen, Germany
[12]Institute of Latvian History, University of Latvia, Rīga 1050, Latvia

[13]Institut für Ur- und Frühgeschichte, Christian-Albrechts-Universität zu Kiel, Johanna-Mestorf-Straße 2-6, 24118 Kiel, Germany
[14]Lithuanian Institute of History, Kražių st. 5, Vilnius 01108, Lithuania
[15]Museum Lolland-Falster, Frisegade 40, 4800 Nykøbing Falster, Denmark
[16]Muzeum Podlaskie w Białymstoku, Ratusz, Rynek Kościuszki 10, 15-426 Białystok, Poland
[17]Centre for Baltic and Scandinavian Archaeology (ZBSA), Schleswig-Holstein State Museums Foundation, Schloss Gottorf, Schlossinsel 1, 24837 Schleswig, Germany

BC, 0000-0002-2622-1376; HKR, 0000-0002-4850-692X; AL, 0000-0003-4892-6323; ED, 0000-0003-1843-9620;
EO, 0000-0002-7212-629X; KA, 0000-0002-8847-5670; PMA, 0000-0002-7538-7014; MC, 0000-0002-5953-6550;
AC-Z, 0000-0001-6171-9930; JK, 0000-0002-2118-2005; SK, 0000-0001-7531-0933; HP, 0000-0002-5854-1323;
GP, 0000-0002-4591-8822; SAS, 0000-0003-2685-1527; AT, 0000-0002-5727-7364; MT, 0000-0002-4247-3370;
AW, 0000-0002-2887-387X; JM, 0000-0002-4346-5591; CPH, 0000-0002-5206-7464; OEC, 0000-0002-4296-8402

The introduction of pottery vessels to Europe has long been seen as closely linked with the spread of agriculture and pastoralism from the Near East. The adoption of pottery technology by hunter–gatherers in Northern and Eastern Europe does not fit this paradigm, and its role within these communities is so far unresolved. To investigate the motivations for hunter–gatherer pottery use, here, we present the systematic analysis of the contents of 528 early vessels from the Baltic Sea region, mostly dating to the late 6th–5th millennium cal BC, using molecular and isotopic characterization techniques. The results demonstrate clear sub-regional trends in the use of ceramics by hunter–gatherers; aquatic resources in the Eastern Baltic, non-ruminant animal fats in the Southeastern Baltic, and a more variable use, including ruminant animal products, in the Western Baltic, potentially including dairy. We found surprisingly little evidence for the use of ceramics for non-culinary activities, such as the production of resins. We attribute the emergence of these sub-regional cuisines to the diffusion of new culinary ideas afforded by the adoption of pottery, e.g. cooking and combining foods, but culturally contextualized and influenced by traditional practices.

# 1. Introduction

In Western European archaeological literature, the production of pottery for processing and storing foods has long been regarded as the inevitable consequence of prehistoric farming [1]. Indeed, in most of Europe, early ceramic technology appears in association with the first agricultural and pastoral societies. However, in many other parts of the world, pottery was produced and used by hunter–gatherers [2]. This technological development among hunter–gatherer communities has often been construed as the result of influence from neighbouring farming societies or non-agricultural communities [3]. However, in East Asia, the wider use of scientific dating methods, notably Accelerator Mass Spectrometry (AMS) radiocarbon ($^{14}$C), has completely decoupled pottery use and agriculture, with the former emerging in the Late Pleistocene, many millennia before the introduction of domesticated plants and animals [4]. In Eastern Europe and Western Siberia, the use of pottery among hunter–gatherer societies, established along major river basins (e.g. the Ob', Volga and Don rivers) [5–7], is well attested, and here, in contrast to the Western conception, is regarded as the innovation that defined the start of the Neolithic itself. Indeed, these Eastern 'Neolithic' hunter–gatherer ceramic traditions might have influenced prehistoric hunter–gatherers in present-day Estonia, Latvia, Lithuania and Belarus (including the Narva and Neman cultures) who began producing pottery by the mid-6th millennium cal BC [8–10]. Other evidence of prehistoric pottery use by hunter–gatherers is attested worldwide such as in the Americas and in Africa (e.g. [2,11–13]).

In Northwestern Europe, where the earliest pottery emerged in hunter–gatherer societies during the 5th millennium cal BC (Ertebølle culture in the Southwest Baltic and Swifterbant culture in The Netherlands), the adoption of pottery has been variously attributed to local innovation, cultural transmission from other ceramic-using hunter–gatherers (e.g. [14,15]) or adoption from Neolithic farmers [16]. Indeed, there is evidence of contact, through artefact exchange, between hunter–gatherers of the Baltic and *Linearbandkeramik* (LBK) as well as post-LBK (e.g. Rössen and Stichbandkeramik) farming communities already established to the south and east of the Elbe River [17–19], which may have led to the uptake of ceramics either through direct transmission or through a process of creolization [16].

It is clear that new approaches, both theoretical and methodological, are required to explain the motivations behind the origins and development of ceramic technology [3]. Measuring the degree of shared technological or stylistic features is the obvious approach to track the dispersal of hunter–gatherer pottery. Similarly, AMS dating is required to assess the speed and tempo of dispersal. Here, however, we focus on the functional analysis of hunter–gatherer vessels in order to understand the motivation(s) for

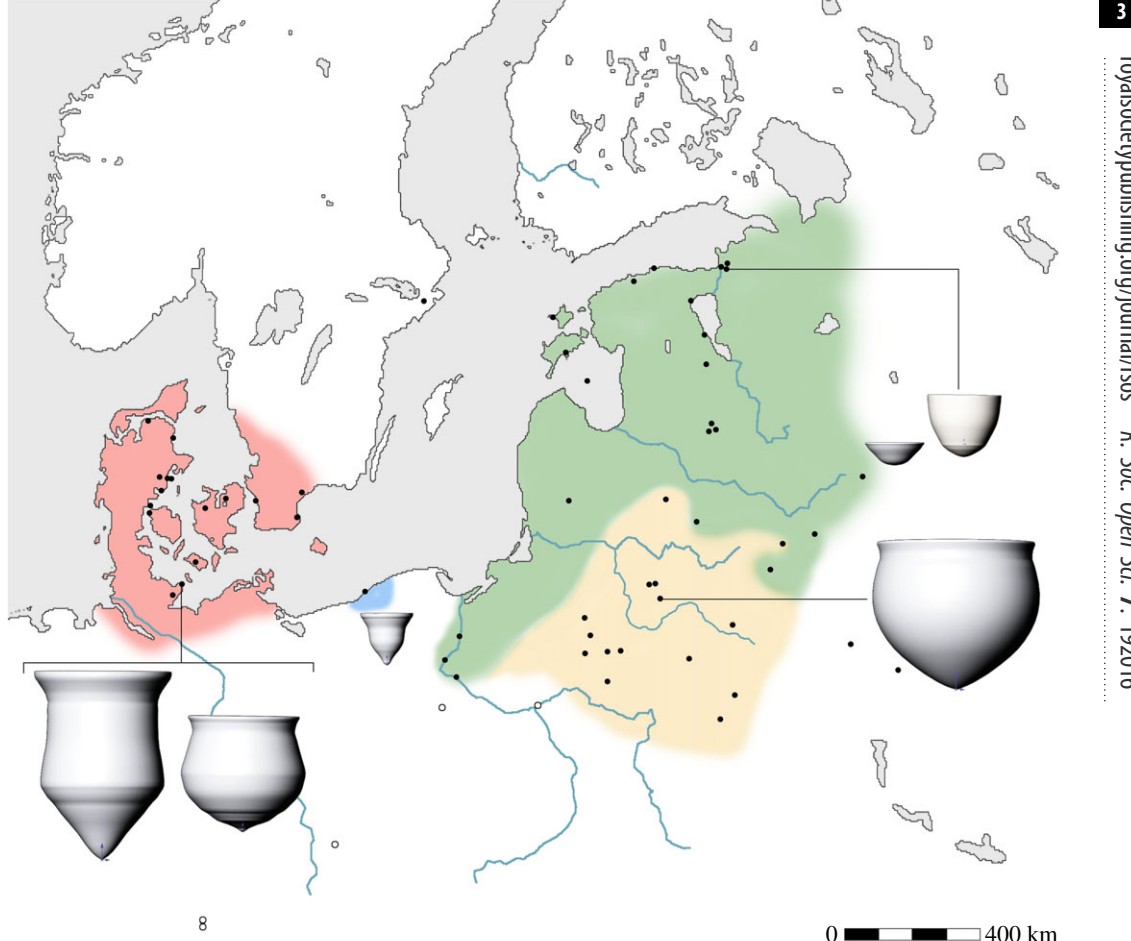

**Figure 1.** Map showing locations of hunter–gatherer (filled circles) and early agricultural sites (open circles) discussed in the text. Also shown is the extent of different hunter–gatherer cultural groups (red, Ertebølle; blue, Dąbki; yellow, Southeastern Baltic and Neman; green, Narva). Individual site names are listed in electronic supplementary material, table S1 and figure S1.

their initial adoption and subsequent use. This is a key question that has received relatively little attention compared to studies of early agriculture pottery use. Focusing on a region at the heart of the debate, the circum-Baltic, here, we explore sub-regional patterns of pottery use by hunter–gatherers in an attempt to directly assess differences and similarities in their need for pottery. A further goal was to trace potential contacts with nearby farming populations that might be manifested through residues derived from domesticated animals or plants. In total, we compared the function of over 500 hunter–gatherer ceramic containers across an E–W transect of the region (figure 1; electronic supplementary material, figure S1) using molecular and isotopic characterization methods to investigate vessel use.

Our study, the largest and most detailed of its kind for the Baltic region, encompasses the earliest Narva ceramics of the Eastern Baltic and Northwest Russia, early pottery in Southern Lithuania, Northern Belarus and Eastern Poland including, Dubičiai, Sokołówek and Neman-Lysaya Gora wares, the Mesolithic ceramics from Dąbki, Poland, and lastly the Ertebølle culture of Denmark, Northern Germany, and Southern Sweden. Although large uncertainties exist regarding when pottery first emerged in these regions, the vessels investigated here mostly date to the late 6th–5th millennium cal BC for the Narva culture in the Eastern Baltic, and the 5th millennium cal BC in the Western Baltic [10,20–22].

Lipids, derived from a range of plant and animal species, accumulate during pottery use and are preserved in the ceramic fabric or within charred surface deposits (foodcrusts or sooted deposits). Individual lipids were separated and their structure determined by Gas Chromatography-Mass Spectrometry (GC-MS). Further information relating to their biological source was gained by measuring their carbon isotope values (Gas Chromatography-Combustion-Isotope Ratio Mass Spectrometry, GC-C-IRMS). This approach has been widely applied to European Neolithic pottery [22–29], and is increasingly being applied to hunter–gatherer pottery throughout the circum-Baltic region [22,30–32]. Here, we report the outcome of organic residue analysis undertaken on 416 new

samples which, with previously reported data, constitute a set of 667 analyses, including 467 ceramic sherds and 200 charred surface deposits from 528 pottery vessels from hunter–gatherer sites across the study region (electronic supplementary material, dataset 1).

# 2. Material and methods

## 2.1. Materials

The results undertaken on 416 samples (ceramic potsherds as well as interior and exterior charred surface deposits) are presented here for the first time, complementing existing data on 251 samples available in the literature (electronic supplementary material, table S1, dataset 1). This represents 528 cooking vessels from 61 early pottery sites located in the circum-Baltic region from the late 6th to the beginning of the 4th millennium cal BC (figure 1; electronic supplementary material, figures S1 and S2). The results obtained on shallow elongated oval bowls, assumed to be lamps, from Narva and Ertebølle cultural contexts as well as at Dąbki, were deliberately excluded from the study as their use is probably not associated with food preparation and/or consumption [33]. Archaeological sites associated with the earliest stages of pottery formed the majority of the analysed samples.

In the Western Baltic region, pottery was adopted at Grube-Rosenhof at *ca* 4600 cal BC making this coastal site one of the earliest yielding Ertebølle pottery [34]. An extensive study of this assemblage (59 vessels) was undertaken alongside the analyses of pottery (188 vessels) from the Ertebølle sites of Bjørnsholm, Flynderhage, Gamborg Fjord, Havnø, Hjarnø, Neustadt, Ringkloster, Ronæs Skov, Stenø, Syltholm (Rødbyhavn; MLF906-I/906-II/939-I), Tybrind Vig, Åkonge and Åle (some data reported in [22,32,35,36]). While we targeted pottery from some of the earliest Ertebølle sites in the region (e.g. Grube-Rosenhof), we also sampled Ertebølle vessels from younger sites. These data were supplemented by 60 pointed-based vessels sampled from the site of Dąbki (site 9) located on the Pomeranian coast, Poland. Occupied from *ca* 5200–3700 cal BC, the long depositional sequence includes locally made Mesolithic pottery and Early Neolithic Funnel Beakers (TRB), as well as pottery imports from farming societies to the south, predominantly dating to the later 5th millennium cal BC [37,38]. This unique ceramic assemblage makes Dąbki key for understanding dispersal patterns and cultural exchange between hunter–gatherers of the Baltic region and neighbouring LBK and post-LBK farmers as well as far-reaching contacts with the Hungarian Plain [39]. The proposed appearance of Mesolithic pottery at Dąbki as early as *ca* 4850/4700 cal BC [40] is difficult to sustain, given the results of organic residue analysis presented in this paper, which shows that most of the foodcrusts at Dąbki are composed of carbon from an aquatic source and therefore subject to a significant reservoir effect; most, if not all of the material sampled may date to the second half of the 5th millennium BC.

Early pottery found in the Southeastern Baltic and adjacent regions (Southern Lithuania, Northern Belarus and Eastern Poland) has not been investigated previously using organic residue analysis. Consequently, knowledge regarding the function and use of these ceramic vessels remains limited, and the material dated up to now is extremely scarce. For the present research, vessels of the Dubičiai type (18 vessels), defined as the earliest stage of the Neman culture or as a separate cultural type predating the Neman culture [41,42], of the Sokołówek type (six vessels), chronologically and culturally very close to Dubičiai [43,44], and of the Lysaya Gora phase (middle stage of the Neman culture, 15 vessels) were sampled. In addition, Early Neolithic pottery vessels from Northern Belarus were analysed (nine vessels). They represent other possibly synchronous or slightly older cultural traditions dating back to at least the end of the 6th millennium cal BC as demonstrated by a new AMS radiocarbon ($^{14}$C) date obtained on a charred surface residue from Lučyn Barok Siamionaŭski ($6152 \pm 28$ BP; OxA-38706; electronic supplementary material, figure S3), and which may be connected with a wider early ceramic distribution area further to the east [45]. All these vessels were derived from a total of 18 sites located in Northern Belarus (Drazdy 12, Dubovy Loh 5, Kamen' 6, Lučyn Barok Siamionaŭski, Rusakova, Sien'čycy 3), Northeast Poland (Bransk-site 22, Grądy-Woniecko, site 1, Jeroniki, Krzemienne, Sośnia-site 1, Stacze-site 1) and Southern Lithuania (Dubičiai 3, Glūkas 3, Gribaša 4, Jara 2, Karaviškės 6, Varėnė 10). To address the issue of an overall lack of $^{14}$C dates for the cultural groups in question, ceramics of the early phases were selected based, mostly, on their technological features (type of paste recipes, modelling technique and surface treatment), and by their stratigraphic context (where possible). Consequently, only ceramic fragments that were attributed with a high degree of certainty to a specific pottery stage were considered for the present study.

Moreover, Narva pottery from 24 sites located in Northern Belarus, Estonia, Latvia, Lithuania, Northeast Poland and Northwest Russia were sampled and are considered as a whole for the first time (some data reported in [30,46]). Since Narva pottery from present-day Estonia has already been extensively studied [30], a particular emphasis was placed on material from sites in Latvia (Iča, Osa, Zvidze) and Lithuania (Daktariškė 5, Kretuonas 1). Because the sites of Osa (24 vessels) and Zvidze (31 vessels) have yielded the oldest direct dates on Narva pottery (ca 5500 cal BC, see electronic supplementary material, figure S3 and comments) [20,21] in the region, they were of particular interest in order to understand the dynamics behind the emergence of pottery in the Eastern Baltic. In addition, pottery recovered from sites located on the margin of the Narva culture were included (Serteya X, XIV, Rudnya Serteyskaya in Northwest Russia and Asaviec 4, Biarešča 4, Zacennie in Northern Belarus). These data were supplemented by five imported or imitation Narva vessels from three sites in Eastern Poland (Kaldus, site 3, Sasieczno, site 4 and Welcz Wielki, site 10A).

## 2.2. Lipid extraction

Prior to lipid extraction, the external layer of each potsherd was removed in order to reduce the potential for contamination from the burial environment or during post-excavation. Moreover, a method blank and standard were extracted alongside each batch of samples to identify any contaminants introduced during the extraction. The direct *in situ* transesterification, or acidified methanol, extraction method was applied to 1 g of ceramic powder and *ca* 20 mg of charred surface deposits following a routine procedure [47]. Briefly, methanol and sulfuric acid were added to the sample prior to a heating step (70°C, 4 h), three successive extractions with *n*-hexane and neutralization with potassium carbonate. The hexane phases were combined into vials, and copper turnings added in order to remove cyclic octaatomic sulfur that was present in many of the samples. Finally, the extracts were dissolved in hexane and analysed by GC-MS. Internal standards (10 µg, alkanes $C_{34}$ and $C_{36}$) were added at the beginning and end of the extraction procedure, respectively.

Solvent extraction was undertaken on 12 selected samples from Grube-Rosenhof and Neustadt to investigate either the presence and distribution of triacylglycerols (TAGs) or the presence of other intact lipids (e.g. wax esters). The samples were selected since they either had $\Delta^{13}C$ ($C_{18:0}$-$C_{16:0}$) values indicating the presence of dairy products (see Results and discussion) or had significant quantities of *n*-alkyl lipids in the acidified methanol extract indicating the potential presence of plant products or beeswax. Dichloromethane-methanol (4 ml) mixture (2 : 1, v/v) was added to 1 g of ceramic powder. Then, the solution was ultrasonicated for 15 min and centrifuged for 10 min. The supernatant was transferred to a clean tube. Two additional solvent extraction steps were performed and the three extracts were combined. When required, a filtration step with glass wool was added and the samples were dried under $N_2$. Prior to GC-MS analysis, 100 µl of BSFTA-1% TMCS was added to the lipid extracts, which were heated at 70°C for 1 h.

## 2.3. Gas Chromatography-Mass Spectrometry

GC-MS analysis of the acidified methanol extracts was undertaken using an Agilent 7890B series gas chromatograph coupled to an Agilent 5977B mass spectrometer equipped with a quadrupole mass analyser (Agilent Technologies, Cheadle, Cheshire, UK). A split/splitless injector (used in splitless mode) was maintained at 300°C and a HP-5MS or DB-5MS column (30 m × 250 µm × 0.25 µm; Agilent Technologies, Cheadle, Cheshire, UK) was used. Helium was the carrier gas (3 ml min$^{-1}$). The oven temperature was set at 50°C for 2 min, then raised by 10°C min$^{-1}$ until 325°C was reached, where it was held for 15 min. The ionization energy of the mass spectrometer was 70 eV and spectra were obtained in scanning mode between *m/z* 50 and 800.

Subsequently, the lipid extracts were analysed using a GC-MS equipped with a DB-23 (50%-cyanopropyl)-methylpolysiloxane column (60 m × 0.250 mm × 0.25 µm; J&W Scientific, Folsom, CA, USA). The oven temperature was set at 50°C for 2 min before increasing to 100°C (10°C min$^{-1}$). The temperature was then raised by 4°C min$^{-1}$ to 140°C, then by 0.5°C min$^{-1}$ to 160°C and, finally, by 20°C min$^{-1}$ to 250°C where it was maintained for 10 min. The SIM (selected ion monitoring) mode was used in order to target the specific markers of aquatic resources using characteristic ions groups: *m/z* 74, 87, 213, 270 for 4,8,12-trimethyltridecanoic acid (TMTD), *m/z* 74, 88, 101, 312 for pristanic acid, *m/z* 74, 101, 171, 326 for phytanic acid and *m/z* 74, 105, 262, 290, 318, 346 for the detection of ω-(*o*-alkylphenyl)alkanoic acids of carbon lengths $C_{16}$ to $C_{22}$ ($APAA_{16-22}$). In addition, separation of the two phytanic acid diastereomers (3S,7R,11R,15-phytanic acid or SRR and 3R,7R,11R,15-phytanic acid

or RRR) was obtained, which enabled the calculation of the percentage of SRR in total phytanic acid (SRR %) by integrating the $m/z$ 101 ion [48]. The carrier gas used was helium with a flow rate of 1.5 ml min$^{-1}$.

GC-MS analysis of the solvent extracts was performed using an Agilent 6890 N series gas chromatograph coupled to an Agilent 5975C mass spectrometer equipped with a quadrupole mass analyser and EI source (Agilent Technologies, Santa Clara, CA, USA). The solvent extracts (1 µl) were injected into a PTV injector with on column adaptor connected to an SGE HT-5 column (12 m × 0.22 mm × 0.1 µm; Tragan, Ringwood, Victoria, Australia). Helium was used as the carrier gas (1.1 ml min$^{-1}$). Separation was conducted using a GC oven temperature programme consisting of an initial temperature of 50°C, which was held for 2 min, followed by a 10°C min$^{-1}$ ramp to 370°C; the oven was held at 370°C for a further 15 min. Spectra were obtained in full scan mode ($m/z$ 50 to 1000).

## 2.4. Gas Chromatography-Combustion-Isotope Ratio Mass Spectrometry

GC-C-IRMS measurements of the main fatty acid methyl esters ($C_{16:0}$ and $C_{18:0}$) were undertaken using a Delta V Advantage isotope ratio mass spectrometer (Thermo Fisher, Bremen, Germany) linked to a Trace Ultra gas chromatograph (Thermo Fisher) with a GC Isolink II interface (Cu/Ni combustion reactor held at 1000°C; or CuO combustion reactor held at 850°C). Ultra high-purity-grade helium with a flow rate of 2 ml min$^{-1}$ was used as the carrier gas, and parallel acquisition of the molecular data was achieved by deriving a small part of the flow to an ISQ mass spectrometer (Thermo Fisher). Hexane was used to dilute the samples, and 1 µl of each sample was injected into a DB-5MS ultra-inert fused-silica column (60 m × 0.25 mm × 0.25 µm; J&W Scientific). The temperature was set at 50°C for 0.5 min and raised by 25°C min$^{-1}$ to 175°C, then raised by 8°C min$^{-1}$ to 325°C, where it was held for 20 min. A clear resolution and a baseline separation of the analysed peaks was achieved.

Eluted products were combusted to $CO_2$ and ionized in the mass spectrometer by electron impact. Ion intensities of $m/z$ 44, 45 and 46 were monitored in order to automatically compute the $^{13}C/^{12}C$ ratio of each peak in the extracts. Computations were made with Isodat (v. 3.0; Thermo Fisher) and IonOS Software (Isoprime, Cheadle, UK), and were based on comparisons with a repeatedly measured standard reference gas ($CO_2$). The results from the analysis are reported in parts per mil (‰) relative to an international standard (V-PDB). Each batch of samples was calibrated using a calibration curve (average $R^2 = 0.996 \pm 0.007$ in 32 batches) based on expected versus measured $\delta^{13}C$ values of $n$-alkanes and $n$-alkanoic acid esters international standards (Indiana A6 and F8-3 mixture). More precisely, the accuracy of the instrument was determined on $n$-alkanoic acid ester standards of known isotopic composition (Indiana standard F8-3, 76 measurements). The mean ± s.d. values of these were −29.93 ± 0.11‰ and −23.22 ± 0.08‰ for the methyl ester of $C_{16:0}$ (reported mean value versus V-PDB −29.90 ± 0.03‰) and $C_{18:0}$ (reported mean value versus V-PDB −23.24 ± 0.01‰), respectively. Precision was determined on a laboratory standard mixture that was injected regularly between samples (426 measurements). The mean ± s.d. values of $n$-alkanoic acid esters were −30.79 ± 0.16‰ for the methyl ester of $C_{16:0}$ and −26.31 ± 0.23‰ for the methyl ester of $C_{18:0}$. Values were corrected subsequent to analysis to account for the methylation of the carboxyl group, using a mass balance formula. Corrections were based on comparisons with a standard mixture of $C_{16:0}$ and $C_{18:0}$ fatty acids of known isotopic composition, processed in each batch under identical conditions. Each sample was measured at least twice, while the standard deviation provided takes into account the propagation of uncertainties between the replicate measurement of: (i) the sample, (ii) the methylated standard and (iii) the $C_{16:0}$ and $C_{18:0}$ fatty acid standard measured offline.

## 2.5. Mixing models (FRUITS)

A Bayesian approach was adopted in order to evaluate the probability of mixing different food resources in the investigated hunter–gatherer pottery as well as published LBK pottery [49–52]. This was achieved using the 3.0 Beta version (http://sourceforge.net/projects/fruits/) of the Bayesian mixing model FRUITS [53] by applying concentration-dependent models that used $\delta^{13}C_{16:0}$ and $\delta^{13}C_{18:0}$ values as proxies (see electronic supplementary material, table S2). The first model (Model A) considered four potential food resources (ruminant, porcine, marine and freshwater) established from the $\delta^{13}C$ values of new (electronic supplementary material, table S3, dataset 2) and published authentic reference animal tissues [22,24,48,54–58]. Uncertainties were derived using a covariance matrix and standard errors of the mean $\delta^{13}C$ values for each food source, assuming a multivariate normal distribution and that the vessels were used repeatedly. Palmitic and stearic acid concentration values were based on the USDA Food Composition Databases (electronic supplementary material, table S3) with uncertainties expressed by

their standard deviations. The concentrations and model outputs are expressed as percentage of total lipid by weight. Numerical Bayesian inference was performed using the BUGS software, a Markov chain Monte Carlo (MCMC) method that employs Gibbs sampling and the Metropolis–Hastings algorithm. The first 5000 iterations of the MCMC chains were discarded (burn-in steps) and these were then run for an additional 10 000 iterations. Model convergence was checked by inspecting if the trace plots of the respective posterior chains exhibited an asymptotic behaviour. Trace autocorrelation plots were also inspected to assess convergence. A second model (Model B), including five resources (ruminant adipose, ruminant dairy, porcine, marine and freshwater), was applied to 15 samples given their large isotopic discrepancy between the main fatty acids characteristic of ruminants ($\Delta^{13}C < -4.3‰$) using the same parameters as above.

## 2.6. Spatial modelling of the estimated contribution of different food resources to pottery

A local average model (AverageR) available in the IsoMemo app v. 1.4.6 (https://www.isomemoapp. com/), developed within the IsoMemo and Pandora initiative, was used to generate spatial estimates of the contribution of aquatic products, ruminant and porcine fats based on the FRUITS outputs (described above). AverageR is a generalized additive mixed model that uses a thin plate regression spline [59]. The model used the average percentage contribution of each product to the total lipid in each vessel, and its respective standard deviation. In the case of freshwater and marine resources, these values were summed in order to evaluate the total contribution of aquatic resources.

# 3. Results and discussion

## 3.1. Evidence for the processing of aquatic resources

Overall, 84% of the samples yielded interpretable amounts of lipids (i.e. greater than 5 µg g$^{-1}$ for potsherds and greater than 100 µg g$^{-1}$ for charred surface deposits; [60]; electronic supplementary material, dataset 1), with lipids particularly well preserved in charred surface deposits especially those from waterlogged sites (median$_{FC}$ = 1072 µg g$^{-1}$, median$_{FC\text{-waterlogged sites}}$ = 1371 µg g$^{-1}$, median$_{sherds}$ = 50 µg g$^{-1}$). Many of the vessels had clear evidence for the processing of aquatic resources, i.e. fish, birds, aquatic mammals (seals as well as beaver), and/or shellfish. Distinctive arrays of ω-(o-alkylphenyl)alkanoic acids (APAAs), with 18, 20 and/or 22 carbon atoms, produced by heat alteration of polyunsaturated fatty acids in tissues of these animals [61,62], were present in a substantial proportion of the majority of the samples: Dąbki (66%), Ertebølle (55%) and Narva (50%). Other compounds consistent with aquatic resources, including isoprenoid fatty acids, were frequently identified (electronic supplementary material, dataset 1). The ratio of the two diastereomers (SRR/RRR) of phytanic acid provided a confirmatory proxy [48] for the presence of aquatic foods in many of the analysed samples.

   Similar frequencies of compounds derived from aquatic organisms have been reported in other hunter–gatherer vessels from East Asia, North America and the Arctic [12,47,63–67] but are virtually absent in agricultural Neolithic pottery [24,27,29], with the exception of some sites that have earlier Mesolithic occupation [22,26,68]. APAAs were particularly prevalent in charred surface deposits (92% contained at least one APAA, electronic supplementary material, figure S4), most likely through protracted heating. An alternative explanation is that charred surface deposits form more readily when cooking tissues with a high polyunsaturated fatty acid content, such as fish or aquatic mammal oils. Interestingly, APAAs were almost absent in the early pottery of the Southeastern Baltic region, which also had very few charred surface deposits ($n = 1$). While the presence of aquatic biomarkers across all assemblages analysed was comparatively high, their absence is difficult to interpret but could indicate either differences in use or, most likely, conditions less favourable for preservation.

   To investigate differences across the region further, we examined the stable carbon isotope values of the individual mid-chain length fatty acids, stearic (C$_{16:0}$) and palmitic (C$_{18:0}$) acid, extracted from the samples (figure 2). As most of the samples yielded high quantities of these compounds, this approach avoids biases associated with differential formation and preservation of biomarkers permitting a more robust statistical approach. Striking contrasts between cultural groups were apparent when the stable carbon isotope ($\delta^{13}C$) value of C$_{16:0}$ was plotted against the value for C$_{18:0}$ (figure 2). Much of this isotopic variation, however, can be explained by the location of the sites in different ecological settings. Lipids extracted from pottery from inland and estuarine/lagoonal hunter–gatherer sites (Dąbki, majority of the Narva sites) are depleted in $^{13}C$ compared to those from coastal sites (majority

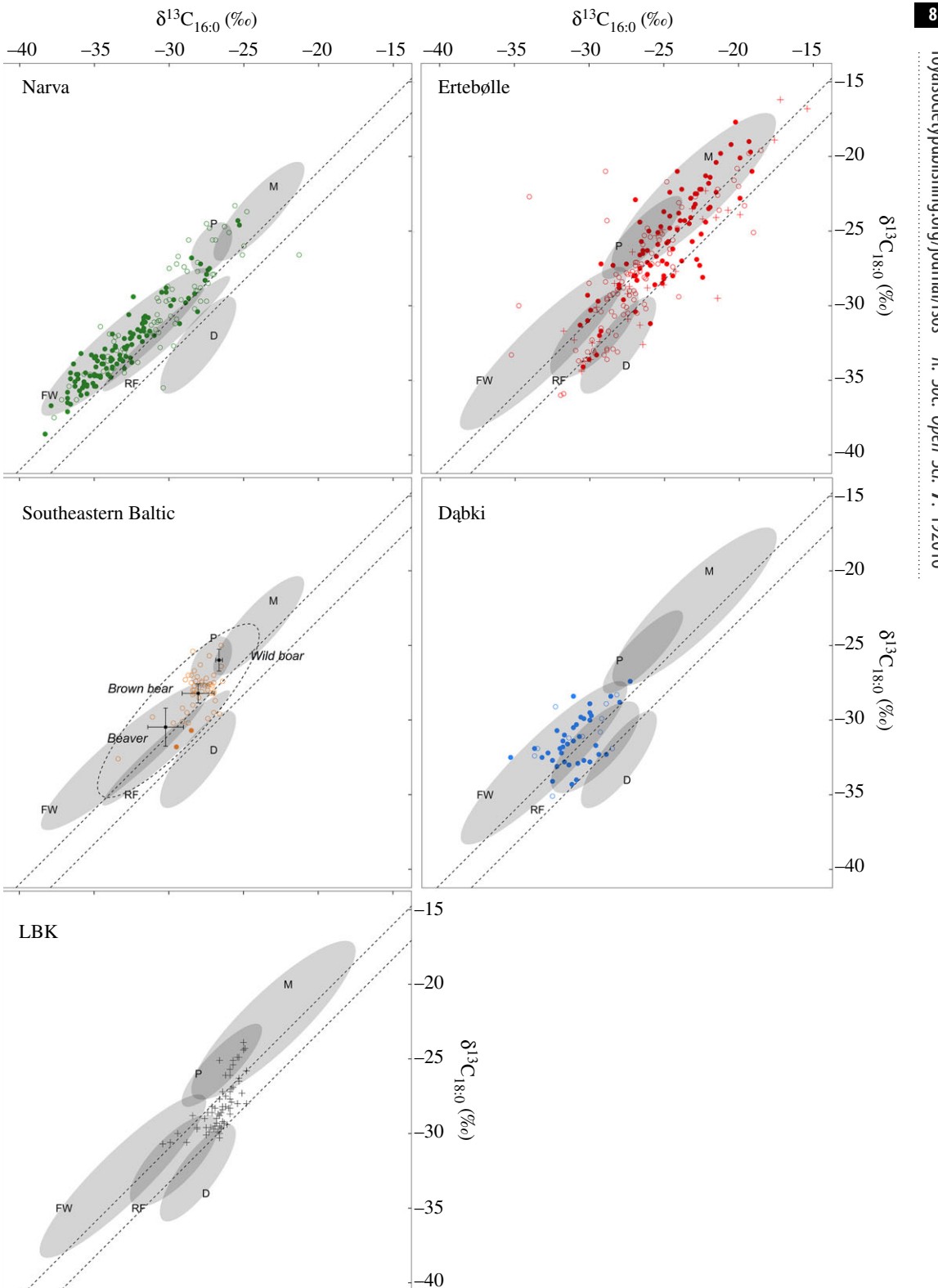

**Figure 2.** δ¹³C values of C$_{16:0}$ and C$_{18:0}$ fatty acids from circum-Baltic hunter–gatherer pottery belonging to the Narva, Ertebølle and early pottery cultures of the Southeastern Baltic, as well as the Mesolithic ceramic culture at Dąbki. Published values from early agricultural LBK sites are presented for comparison. The presence of APAA C$_{20}$ (aquatic biomarker) is indicated by a filled circle, while its absence is noted by an open circle (cross when it is undetermined). Ninety-five per cent confidence ellipses were created taking into account values obtained from modern animals [22,24,48,54–58] (electronic supplementary material, dataset 3). M, marine products; FW, freshwater resources; P, porcine; RF, ruminant adipose; D, dairy products.

of the Ertebølle sites), consistent with a freshwater (and/or terrestrial) rather than marine origin. At the coastal sites, fatty acid $\delta^{13}C$ values are positively correlated with the phytanic acid SRR/RRR ratio ($R_{coastal} = 0.6$; electronic supplementary material, figure S5) and the presence of APAAs, while a negative correlation between these criteria was observed at the inland sites ($R = -0.3$; electronic supplementary material, figure S5). These independent proxies provide compelling evidence that aquatic organisms were widely processed in hunter–gatherer pottery from the region, especially in Narva pottery, although variation in the isotopic data show that in some cases they were mixed with the fats and oils from other, terrestrial products. Numerous fish remains (electronic supplementary material, table S4; [69–72]), a variety of fishing implements (e.g. weirs, traps, spears, leisters, hooks, nets; [34,73,74]) as well as several hundred coastal sites, including in the case of the Ertebølle culture, shell middens [75,76], corroborate the prominent role of aquatic resources within Ertebølle and Narva communities. In that context, pottery is likely to have constituted a valuable asset for processing and managing abundant aquatic resources.

## 3.2. Pottery used to process ruminant animal products in the Western Baltic

At Dąbki and the Ertebølle sites, a significant proportion (29% and 50%, respectively) of the samples had relatively lower $\delta^{13}C_{18:0}$ compared to $\delta^{13}C_{16:0}$ values (i.e. $\Delta^{13}C$ lower than $-1‰$; electronic supplementary material, figure S6), a well-established proxy reflecting the unique metabolic physiology of ruminant animals [77–79]. But, recent research suggests that the utility of this proxy is compromised in regions where marine/$C_4$ and terrestrial $C_3$ foods are mixed, especially when the fatty acid concentrations vary between food sources [80]. To quantify the potential food source contributions more accurately, we used a mixing model (Model A) that takes into account the relative concentration of $C_{16:0}$ and $C_{18:0}$ in each source as well as their respective $\delta^{13}C$ values and associated uncertainties [53,81]. Data on source food values were obtained from modern authentic fats and oils ($n = 269$, Western Baltic, $n = 135$, Eastern Baltic, electronic supplementary material, dataset 2) with an appropriate correction for the Suess effect [82].

In agreement with the GC-MS data, the summed density distributions obtained from the mixing model (Model A, electronic supplementary material, dataset 3) show that freshwater resources are a major contributor to the residues in Narva pottery as well as at the site of Dąbki, while marine resources predominated in Ertebølle pottery, consistent with their general, coastal location. Despite a high degree of equifinality, a higher percentage of ruminant products is noticeable at Dąbki (figures 3 and 4) while the isotope data show that Ertebølle vessels were used for processing a wider range of foods, including meat from terrestrial animals. These clear differences in the use of pottery may reflect distinct cultural traditions and practices, ranging from a dedicated technology for processing aquatic foods in the Eastern Baltic to a more general, multi-source food processing technology in the Central and Western Baltic sites [32]. This variability cannot be attributed to broader differences in the subsistence economy of Baltic hunter–gatherers. Comparing faunal assemblages is hampered by inter-regional differences in preservation and excavation methods but on a general level it is clear that wild ruminants were commonly hunted throughout the Baltic region (electronic supplementary material, table S4; [70,71]). In the Eastern Baltic, bones of aurochs (*Bos primigenius*), deer (*Cervus elaphus* and *Capreolus capreolus*) and elk (*Alces alces*) are commonly recovered. Elk, for instance, are known to have been hunted *en masse* both by ceramic-using and earlier aceramic hunter–gatherers of this sub-region (e.g. [71]). Yet, there is very little evidence of these animal products in the pottery residues from the Eastern Baltic even accounting for mixing. Conversely, we found ruminant fats in pottery recovered from coastal Ertebølle sites (e.g. Grube-Rosenhof, Neustadt and Tybrind Vig), including shell middens, composed predominantly of marine taxa (e.g. Frederiksodde and Havnø).

Out of the entire sample set ($n = 528$), 15 vessels have $\Delta^{13}C$ values that fall below the limit for wild ruminant carcass fats ($-4.3‰$; [54]). These vessels, the majority ($n = 13$) from Ertebølle sites, meet the widely accepted criteria for prehistoric dairy fats [27,77,83]. As the presence of domesticated animals, apart from dogs, at Ertebølle sites is disputed [84], these data must be examined critically. Theoretically, $\Delta^{13}C$ values within the dairy range can be produced by mixing aquatic oils with terrestrial products due to the higher relative concentration of $C_{18:0}$ in the latter (electronic supplementary material, figure S7). In order to investigate this issue further, we deployed a separate concentration-dependent mixing model (Model B) that included 'dairy' as a separate group to 'ruminant' and used conservative assumptions regarding dispersal of the source data (Model B, electronic supplementary material, dataset 3). With this model, the number of hunter–gatherer ceramics where dairy credibly made a significant contribution (i.e. where lowest estimation is above

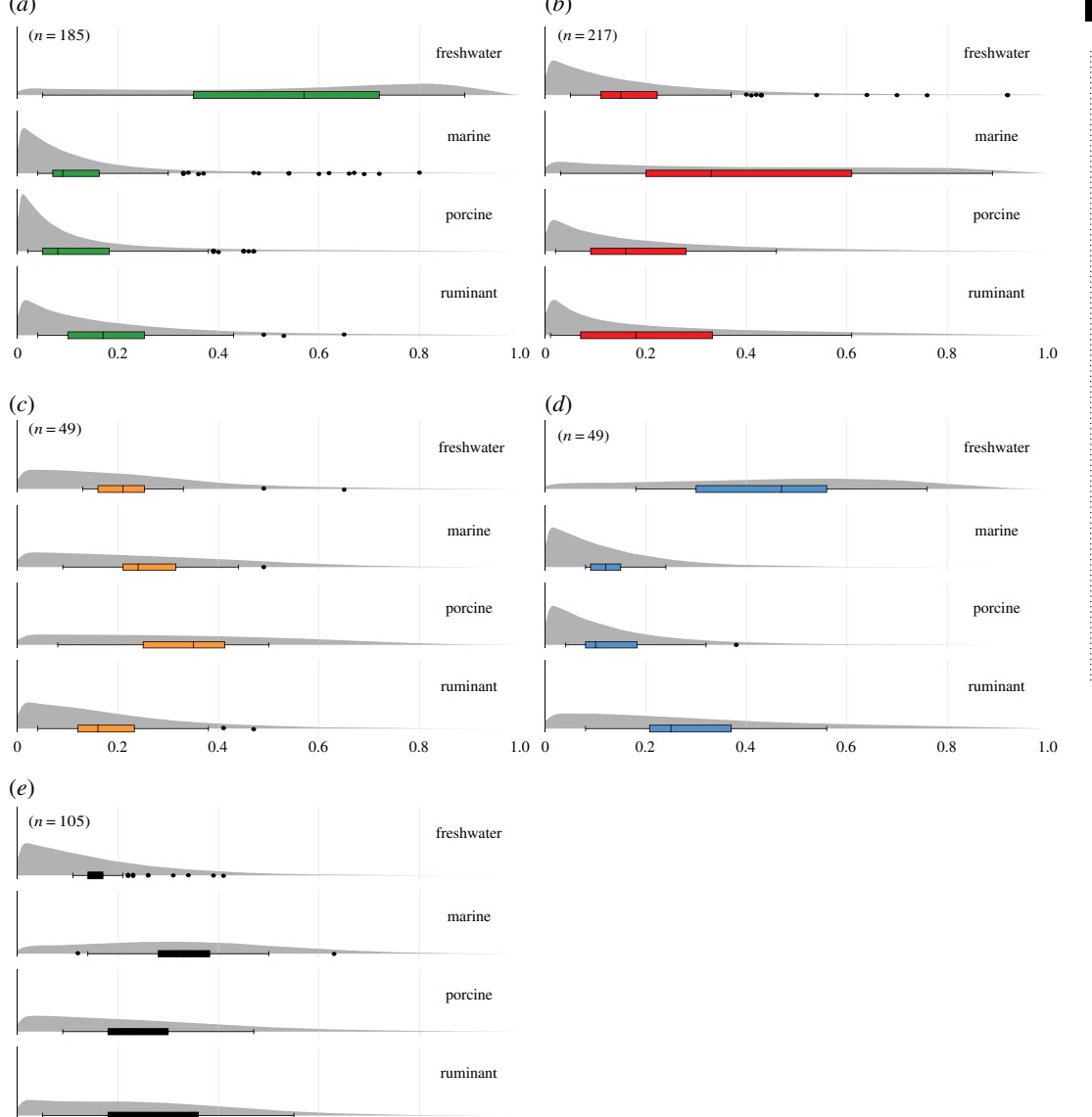

**Figure 3.** Estimated percentage contributions of lipids from different food sources using a concentration-dependent mixing model (Model A): (a) Narva, (b) Ertebølle, (c) Southeastern Baltic and adjacent regions, (d) Dąbki Mesolithic and (e) LBK. Box plots show the range of mean percentage contributions estimated from each pot for each food source. The model parameters are described in the electronic supplementary material. The summed probability density distributions (grey) show the relative likelihood of the contribution of each food resource summed across the sample groups and normalized to account for differences in sample size. Brackets denote the number of samples analysed.

5% of the total fatty acid) is reduced significantly. Nevertheless, for five hunter–gatherer vessels, four securely assigned to the Ertebølle culture (2 × Grube-Rosenhof, 2 × Neustadt) and one as an imported Narva sherd or imitation (Kaldus, electronic supplementary material figure S8), the probability density distributions show that dairy foods cannot be ruled out considering the lower bounds of the 95% credible interval (electronic supplementary material, table S5). None of these vessels contained molecular evidence of aquatic oils (electronic supplementary material, dataset 1). The poor preservation of acyl lipids in these vessels, however, precludes supporting molecular evidence of dairy products, such as lower molecular weight triacylglycerols [79].

Two hypotheses can be posited to explain the potential presence of dairy products in the Ertebølle vessels. The first assumes the possible management of domesticated animals (at *ca* 4500 cal BC; 5658 ± 23 BP; SUERC-89944; electronic supplementary material, figure S3) well before the generally accepted date for the inception of agriculture and pastoralism some 500 years later. While ancient

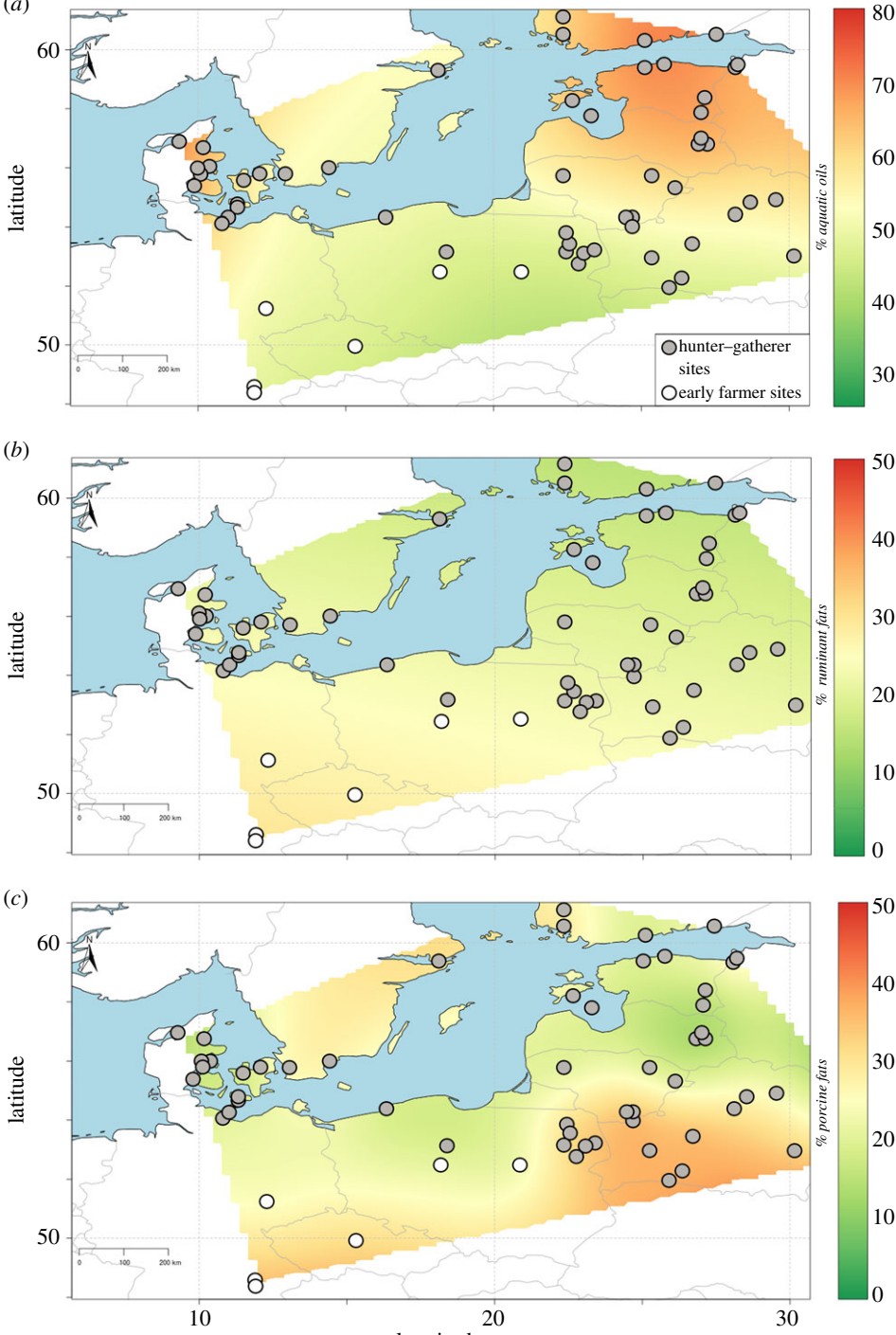

**Figure 4.** Output of AverageR model showing spatial estimates of % product contribution to total fatty acid by weight in hunter–gatherer (filled circle) and farmer (open circle) cooking vessels. (*a*) Aquatic products (marine and freshwater resources); (*b*) ruminant fat (adipose tissue and, potentially, dairy products); and (*c*) porcine fat. Maps created with IsoMemoapp 1.4.3 (see Material and methods).

DNA analysis and morphometrics have suggested that a small number of pig remains from hunter–gatherer sites, including Grube-Rosenhof, are from Near Eastern domesticated stocks [85], and that Mesolithic people managed these animals, their domestic status has been disputed [84,86,87]. In addition, there is no firm osteological evidence for the presence of domesticated ruminant animals prior to *ca* 4000 cal BC [88]. An alternative explanation is that dairy and domestic animal products were exchanged with adjacent LBK and post-LBK farmers for other goods. The Northern German

sites Grube-Rosenhof and Neustadt are located close to the boundary (*ca* 150 km), which divided post-LBK and Ertebølle territories for over 1000 years [89]. Indeed, exchange between hunter–gatherers and farmers in this region is not a new concept [90]. Imported goods such as shoe-last axes are known from many Ertebølle sites [17,34,91,92], while LBK and post-LBK ceramics [92] are present in fewer numbers. If dairy products were exchanged between LBK/post-LBK and Ertebølle hunter–gatherers, the way these were processed most likely differed in the sense that no dairy fats have so far been identified in LBK cooking pots (figure 2; [49–52]), although other ceramic implements (e.g. sieves) have been implicated in the production of fermented dairy products [51].

## 3.3. Non-ruminant terrestrial animals in Southeastern Baltic pottery

The analysis of hunter–gatherer pottery from Southern Lithuania, Northern Belarus and Eastern Poland, including the Neman and other possibly synchronous early ceramic traditions, presents a different use pattern compared to the Ertebølle and Narva pottery. Here, very few aquatic derived lipids were identified, and the $\delta^{13}C$ values of the fatty acids match those of non-ruminant animals (electronic supplementary material, table S6), including wild boar and brown bear (figure 2). It is important to stress that, in more than half of the vessels, pig fat may have contributed to more than 30% (median = 0.34), and up to 50% of the fatty acid content (figures 2 and 3). Associated faunal remains are rare due to the acidic nature of the soils where the sites are located. Nevertheless, wild boar, elk and red deer have been routinely identified in Mesolithic faunal assemblages throughout the region [93], thus it seems reasonable to assume that similar species were present at the time when early pottery groups emerged. More recently, the consumption of terrestrial game was elucidated by the isotopic compositions of charred surface deposits adhering to Neman-type pottery [41,94]. Yet such studies are still very limited for ceramic-using hunter–gatherers in the Southeastern Baltic and adjacent areas. Although the results need to be cautiously interpreted due to the small number of vessels per site that have been investigated, our residue data provide a first glimpse into the cuisine and consumption practices of the region.

## 3.4. Evidence for plant and insect products

The presence of plants in pottery is difficult to assess, due to the fact that they generally have a lower lipid content compared to animal products, often lack specific biomarkers and are characterized by a broad range of isotopic values [7,95,96]. For this reason, the contribution of plant resources to circum-Baltic pottery cannot be excluded. Indeed, plant-derived compounds including pentacyclic triterpenes (e.g. amyrin derivatives), plant sterols (β-sitosterol) and long-chain odd-numbered alkanes (including the predominant *n*-nonacosane) were identified in some samples (electronic supplementary material, dataset 1). Several vessels from Grube-Rosenhof were characterized by long-chain even-numbered fatty acids and alcohols indicative of degraded plant waxes. Some of these do not necessarily derive from the processing of edible plants. Lupane and abietic acid derivatives, commonly regarded as molecular proxies for birch (*Betulus* sp.) and pine (*Pinus* sp.) products (e.g. tar and pitch), were detected in several samples. The molecular approach deployed, unfortunately, does not enable taxonomic identification; consequently we can only speculate on the types of plants processed, which based on archaeobotanical evidence and complementary micro- and macroscopic analysis is very broad (e.g. grasses (*Gramineae*), plantains (*Plantago lanceolata* and *Plantago major*), hazelnuts (*Corylus* sp.), acorns (*Quercus* sp.), mistletoe (Santalaceae) wild berries and spices among others; [97–100]). Furthermore, the presence of long-chain palmitic wax esters alongside *n*-alkyl lipids, which are characteristic of beeswax [101]), were identified in one vessel from Grube-Rosenhof (ROS 8.32-I; electronic supplementary material, figure S9) supplementing existing evidence of beeswax in Ertebølle pottery [102].

## 4. Conclusion

Explaining the uptake of pottery by hunter–gatherers globally is not straightforward and overall we show that the motivations are more varied and complex than previously thought. In the circum-Baltic, organic residue analysis confirms that food preparation was indeed the main function of vessels designated as 'cooking pots'. Mixtures of foodstuffs were identified in the majority of vessels. Resins, tars and pitches, as well as beeswax, were comparatively rare despite the fact that these products are readily identifiable in archaeological pottery [101,103]. Across the region, pots of variable sizes were

produced in large numbers and deposited alongside domestic food waste, implying that they mainly had a utilitarian function. The presence of charred surface deposits and heat transformation products in many of these vessels suggests that cooking was the principal techno-functional driver for their adoption. Remarkably, however, analysis of their contents revealed broad trends between sub-regions and cultural groupings; aquatic resources in the Eastern Baltic, non-ruminant animal fats in the Southeastern Baltic, and a more variable use, including ruminant animal products and plants, in the Western Baltic.

We argue that the observed sub-regional variation in pottery use cannot simply be explained by differences in the environmental settings and resource availability. For example, while faunal assemblages from Narva culture sites demonstrate that both terrestrial and aquatic ecotones were exploited [71,104–106], pottery appears to have been almost exclusively used for the processing of aquatic resources. Instead hunter–gatherer pottery use was under strong cultural control. These differences can, therefore, be crudely described as sub-regional 'cuisines'. From an anthropological perspective, this observation is perhaps of no surprise, as all documented hunter–gatherers practice some form of culturally specific custom for food preparation and consumption, often deploying specific material culture for defined tasks [107–109]. Whether such 'culinary traits' can be used to help understand the dispersal dynamics of pottery technology is debatable. Only through detailed analysis of the raw materials and manufacturing techniques can we test whether ceramic form follows function.

Perhaps a more productive interpretive approach is to situate pottery use in a broader culinary historical context that must have included other food preparation methods such as roasting, grilling, drying and fermenting. Mid-Holocene hunter–gatherers were influenced both by their own 'traditional' aceramic culinary practices and through interaction with other ceramic using groups they came into contact with. So while there is scant evidence that the environment or food procurement strategies changed with the advent of pottery, culinary ideas for combining and cooking foodstuffs in ceramic vessels were undoubtedly mutable with adoption motivated by prior beliefs, for example, concerning cooking performance and efficacy or equally notions of novelty and prestige. Following the widespread uptake of ceramic production among aceramic hunter–gatherers, the use of the new technology remained, at a sub-regional scale, strongly influenced by the surrounding foodscape and pre-existing culinary practices.

A good example is the interaction between Ertebølle hunter–gatherers and LBK/post-LBK farmers. Compared to the Eastern Baltic Narva culture and nearly all other hunter–gatherer groups where organic residue analysis has been conducted, the presence of ruminant animal carcass fats in Ertebølle pottery stands out. It is, therefore, tempting to suggest that the tradition of cooking meat in Ertebølle vessels was acquired from agro-pastoralists they came into contact with, but that domesticated ruminant products were substituted by auroch and deer. Similarly, dairy products may have been exchanged with farmers, initiating the first steps toward the Neolithization process in the southwest Baltic [110]. At the same time, the Eastern Baltic tradition of cooking aquatic foods in pottery continued to be practiced by Ertebølle hunter–gatherers. Remarkably this tradition persisted through further disruptive cultural and economic change following the introduction of agriculture at *ca* 4000 cal BC [22]. Therefore, although technologically and economically grounded, cuisine is most usefully viewed as 'dynamic cultural phenomena embedded in social action, worldviews, and social reproduction' [111]. Discerning pottery function, however crudely, provides important insight into how hunter–gatherers valued food, and in turn how they viewed the world around them.

Data accessibility. Additional information pertaining to this article can be found in the electronic supplementary material.

Authors' contributions. C.P.H. and O.E.C. conceived the study. B.C., H.K.R., A.L., E.D., C.P.H. and O.E.C. participated in the design of the study. B.C., H.K.R., A.L., E.D., E.O., C.P.H. and O.E.C. carried out the laboratory work and data analyses. B.C., H.K.R., A.L., E.D., J.M., C.P.H. and O.E.C. drafted the manuscript, with input from all co-authors. All authors gave their final approval for publication.

Competing interests. We declare we have no competing interests.

Funding. This project was supported by the European Research Council (ERC) under the European Union's Horizon 2020 research and innovation programme (grant agreement no. 695539) to C.P.H.; a postdoctoral fellowship from the British Academy to H.K.R.; the Estonian Research Council funding schemes IUT20-7, MOBERC14 and PSG492 to E.O.; and the National Science Centre, Poland to J.K. (grant number: NCN 2017/27/B/HS3/00478).

Acknowledgements. We thank Marise Gorton (University of Bradford) and Matthew von Tersch (University of York) for undertaking the bulk EA-IRMS analyses, and Antony Simpson (The British Museum) for producing the maps.

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
