## [Reviewer comments · Royal Society Open Science]

Review History

RSOS-192016.R0 (Original submission)

Review form: Reviewer 1

Is the manuscript scientifically sound in its present form?

Yes

Are the interpretations and conclusions justified by the results?

Yes

Is the language acceptable?

Yes

Do you have any ethical concerns with this paper?

No

Have you any concerns about statistical analyses in this paper?

No

Recommendation?

Accept with minor revision (please list in comments)

Comments to the Author(s)

Title: is written as an assertion. It should be written better. It is also necessary to specify which region of Europe is being worked on. The work focuses on the different culinary uses of ceramics from different regions of the Baltic. However, it does not conduct an in-depth discussion of the reasons why these different uses exist. The conclusion outlines an idea but does not discuss it in depth.

Keywords: Some of them already appear in the title and others refer to only some of the contexts the article works in (Ertebølle, Narva, Neman). I think they should be changed.

Abstract: it is clear and summarizes the paper very well. However, there is an inconsistency with the chronology mentioned in the text in Materials section. In this last section it is specified that the samples are from the late 6th to the beginning of the 4th millennium cal BC but in abstract it says: mostly dating to the late 6th-5th millennium cal BC. The differences are minimal but should leave no room for doubt.

Introduction: it starts with an introduction to the different positions on the incorporation of ceramics in hunter-gatherer societies. In this regard, there is also literature in both North and South America referring to the use of ceramic containers, fatty acids and hunter-gatherers that the authors could refer to.

The problem of the adoption of ceramics in the different regions of the Baltic is clearly stated. An introduction to the methodology and the sample with which it is intended to address this problem is made.

Materials and method:

First of all, it is not possible to understand why the chronology of Dąbki is unsustainable (page 7 line 17-21).

Secondly, as this paper attempts to discuss the initial use of ceramic technology, an important point is to establish that all samples correspond to similar chronological ranges or that the samples considered correspond to the earliest ceramics. The chronological assignment of the samples is difficult to follow in the text, so it is suggested that a column with the corresponding absolute chronology be incorporated into Table S1.

Although most of the information is published, a minimum specification of the context in which the samples were found would be convenient. All come from excavated sites? There are no surface samples or museum collections? Sometimes the contexts can affect the subsequent chemical analysis of the samples.

The criteria by which 12 samples are selected for TAGs investigation is not specified (page 8 line 46-47).

Finally, all the published samples have the same methodological procedures as those used by the authors for the samples presented here?

Results and discussion:

This section is very well ordered and the conclusions obtained are consistent with the data presented. Although the authors mention the low possibility of recording plant processing, they mention that some samples present plant derived compounds. What are these samples? They are only some of Grube-Rosenhof? How many?

Conclusions:

The authors conclude that the analysis of the content of the ceramics revealed broad trends among the subregions and cultural groupings. They consider that the sub-regional variation

observed in the use of ceramics cannot be explained simply because of differences in environmental settings and resource availability. They affirm that the use of hunter-gatherer ceramics has been strongly cultural control. This last statement is not well articulated. Cultural control can mean many things and it would be appropriate for authors to develop this idea further.

Some ethnographic citations are also required when they state: as all documented hunter-gatherers practice some form of culturally specific custom for food preparation and consumption often deploying specific material cultures for defined tasks.

References:

There are references that are not complete in supplementary material (N^o5).

Review form: Reviewer 2

Is the manuscript scientifically sound in its present form?

Yes

Are the interpretations and conclusions justified by the results?

Yes

Is the language acceptable?

Yes

Do you have any ethical concerns with this paper?

No

Have you any concerns about statistical analyses in this paper?

No

Recommendation?

Accept with minor revision (please list in comments)

Comments to the Author(s)

This is a comprehensive and highly detailed large-scale study using the technique of organic residue analysis to examine hunter-gatherer pottery use in Neolithic Northern and Eastern Europe.

This paper is a valuable contribution to our understanding of diet and subsistence practices across the Baltic region, demonstrating trends in different resource processing among HG groups. The study has situated the results within the context of the broader region and likely cultural context and I am happy to recommend it for publication subject to a few minor comments as detailed below.

Page 5, line 52. I would comment that this is not the 'largest and most detailed' study of its kind and should not be claimed as such. Previous studies have incorporated larger datasets of potsherds e.g. Evershed et al. 2008 and Whelton et al. 2017 where 2200 and 912 sherds, respectively, were analysed. Mukherjee et al. 2008 also analysed 126 surface residues in a large-scale study.

Page 7 paragraph beginning line 24, I would comment that the number of potsherds analysed from the 18 sites sampled in this region is very low (e.g. 1-4 per site) and indeed does not provide a statistically reliable dataset. It is thus very difficult to make meaningful interpretations on pottery from these sites and a caveat should thus be applied.

Page 16, lines 17-19. 'Several vessels from Grube-Rosenhof were characterized by long-chain wax esters and/or ranges of long chain even-numbered fatty acids and alcohols indicative of degraded plant waxes'

I would point out that whilst long chain even-numbered fatty acids and alcohols can be indicative of degraded plant waxes, in combination, the presence of long-chain wax esters is more suggestive of the processing of beeswax rather than plants. There does not seem to be any detailed description of the distributions of such compounds or, indeed, a chromatogram, to make an assessment of this and I suggest the authors either remove this or clarify.

Oddly, on page 16, line 46-48, the authors state that 'Resins, tars and pitches as well as beeswaxes were comparatively rare despite the fact that these products are readily identifiable in archaeological pottery'. This comment seems odd as beeswax has not been mentioned at all in the paper up to that point. See my comments above though where they claim wax esters are indicative of plant processing.

Review form: Reviewer 3

Is the manuscript scientifically sound in its present form?

Yes

Are the interpretations and conclusions justified by the results?

Yes

Is the language acceptable?

Yes

Do you have any ethical concerns with this paper?

No

Have you any concerns about statistical analyses in this paper?

No

Recommendation?

Accept as is

Comments to the Author(s)

The paper presents large original data, and therefore larger analysis and interpretation can be acceptably left for future works. I have no major remarks to make, and the manuscript can be published basically as it is. This being said, the question of temporal resolution and reasoning deserves a comment. Starting from the title, the authors discuss about the 'adoption of pottery' - however, especially the more east you go, the flimsier the temporal framework gets. Despite certain degree of stylistic or technological uniformity, the material may include pottery (of the same type) covering several centuries, even a millennium. Thus, even if the data is in many cases fairly congruous, it's a bold assumption that all analysed pieces would represent the actual 'adoption' phase.

Decision letter (RSOS-192016.R0)

27-Feb-2020

Dear Dr Courel

On behalf of the Editors, I am pleased to inform you that your Manuscript RSOS-192016 entitled "Organic residue analysis shows different regional motivations for the adoption of pottery by European hunter-gatherers" has been accepted for publication in Royal Society Open Science subject to minor revision in accordance with the referee suggestions. Please find the referees' comments at the end of this email.

The reviewers and handling editors have recommended publication, but also suggest some minor revisions to your manuscript. Therefore, I invite you to respond to the comments and revise your manuscript. The main issues to address are: provide more details of the samples included from other studies (Referee #1); and provide more details of some of the lipid analyses (notably long chain even-numbered fatty acids), Referee #2. There are a number of minor points also raised by the referees.

- Ethics statement

- Data accessibility

<http://datadryad.org/submit?journalID=RSOS&manu=RSOS-192016>

- Competing interests

- Authors' contributions

· Acknowledgements

• Funding statement

Because the schedule for publication is very tight, it is a condition of publication that you submit the revised version of your manuscript before 07-Mar-2020. Please note that the revision deadline will expire at 00.00am on this date. If you do not think you will be able to meet this date please let me know immediately.

- 1) A text file of the manuscript (tex, txt, rtf, docx or doc), references, tables (including captions) and figure captions. Do not upload a PDF as your "Main Document";
- 2) A separate electronic file of each figure (EPS or print-quality PDF preferred (either format should be produced directly from original creation package), or original software format);
- 3) Included a 100 word media summary of your paper when requested at submission. Please ensure you have entered correct contact details (email, institution and telephone) in your user account;
- 4) Included the raw data to support the claims made in your paper. You can either include your data as electronic supplementary material or upload to a repository and include the relevant doi within your manuscript. Make sure it is clear in your data accessibility statement how the data can be accessed;
- 5) All supplementary materials accompanying an accepted article will be treated as in their final form. Note that the Royal Society will neither edit nor typeset supplementary material and it will

be hosted as provided. Please ensure that the supplementary material includes the paper details where possible (authors, article title, journal name).

If your manuscript is newly submitted and subsequently accepted for publication, you will be asked to pay the article processing charge, unless you request a waiver and this is approved by Royal Society Publishing. You can find out more about the charges at <https://royalsocietypublishing.org/rsos/charges>. Should you have any queries, please contact openscience@royalsociety.org.

on behalf of Professor Matthew Collins (Associate Editor) and Jon Blundy (Subject Editor)
openscience@royalsociety.org

Associate Editor Comments to Author (Professor Matthew Collins):

Associate Editor: 1

Comments to the Author:

This work aims to evaluate the introduction of ceramic technology in hunter-gatherer societies through the analysis of lipids deposited in pots and charred surface deposits. More than 500 samples from different archaeological sites located in the Baltic region are analysed

Comments from the Editor

Page 11 line 7 "Each sample was measured in replicate and the standard deviation provided take into account the propagation of uncertainties between the replicate measurement of: (i) the sample, (ii) the methylated standard, and (iii) the C16:0 and C18:0 fatty acids standard measured offline."

What does "in replicate mean" can you clarify?

In the same sentence
"standard deviation provided take into"

reads better as
 “standard deviation provided takes into”

Reviewer #1

Title: is written as an assertion. It should be written better. It is also necessary to specify which region of Europe is being worked on. The work focuses on the different culinary uses of ceramics from different regions of the Baltic. However, it does not conduct an in-depth discussion of the reasons why these different uses exist. The conclusion outlines an idea but does not discuss it in depth.

Keywords: Some of them already appear in the title and others refer to only some of the contexts the article works in (Ertebølle, Narva, Neman). I think they should be changed.

Abstract: it is clear and summarizes the paper very well. However, there is an inconsistency with the chronology mentioned in the text in Materials section. In this last section it is specified that the samples are from the late 6th to the beginning of the 4th millennium cal BC but in abstract it says: mostly dating to the late 6th-5th millennium cal BC. The differences are minimal but should leave no room for doubt.

Introduction: it starts with an introduction to the different positions on the incorporation of ceramics in hunter-gatherer societies. In this regard, there is also literature in both North and South America referring to the use of ceramic containers, fatty acids and hunter-gatherers that the authors could refer to.

The problem of the adoption of ceramics in the different regions of the Baltic is clearly stated. An introduction to the methodology and the sample with which it is intended to address this problem is made.

Materials and method:

First of all, it is not possible to understand why the chronology of Dąbki is unsustainable (page 7 line 17-21).

Secondly, as this paper attempts to discuss the initial use of ceramic technology, an important point is to establish that all samples correspond to similar chronological ranges or that the samples considered correspond to the earliest ceramics. The chronological assignment of the samples is difficult to follow in the text, so it is suggested that a column with the corresponding absolute chronology be incorporated into Table S1.

Although most of the information is published, a minimum specification of the context in which the samples were found would be convenient. All come from excavated sites? There are no surface samples or museum collections? Sometimes the contexts can affect the subsequent chemical analysis of the samples.

The criteria by which 12 samples are selected for TAGs investigation is not specified (page 8 line 46-47).

Finally, all the published samples have the same methodological procedures as those used by the authors for the samples presented here?

Results and discussion:

This section is very well ordered and the conclusions obtained are consistent with the data presented. Although the authors mention the low possibility of recording plant processing, they mention that some samples present plant derived compounds. What are these samples? They are only some of Grube-Rosenhof? How many?

Conclusions:

The authors conclude that the analysis of the content of the ceramics revealed broad trends among the subregions and cultural groupings. They consider that the sub-regional variation observed in the use of ceramics cannot be explained simply because of differences in environmental settings and resource availability. They affirm that the use of hunter-gatherer

ceramics has been strongly cultural control. This last statement is not well articulated. Cultural control can mean many things and it would be appropriate for authors to develop this idea further.

Some ethnographic citations are also required when they state: as all documented hunter-gatherers practice some form of culturally specific custom for food preparation and consumption often deploying specific material cultures for defined tasks.

References:

There are references that are not complete in supplementary material (N°5).

Reviewer #2

This is a comprehensive and highly detailed large-scale study using the technique of organic residue analysis to examine hunter-gatherer pottery use in Neolithic Northern and Eastern Europe.

This paper is a valuable contribution to our understanding of diet and subsistence practices across the Baltic region, demonstrating trends in different resource processing among HG groups. The study has situated the results within the context of the broader region and likely cultural context and I am happy to recommend it for publication subject to a few minor comments as detailed below.

Page 5, line 52. I would comment that this is not the 'largest and most detailed' study of its kind and should not be claimed as such. Previous studies have incorporated larger datasets of potsherds e.g. Evershed et al. 2008 and Whelton et al. 2017 where 2200 and 912 sherds, respectively, were analysed. Mukherjee et al. 2008 also analysed 126 surface residues in a large-scale study.

Page 7 paragraph beginning line 24, I would comment that the number of potsherds analysed from the 18 sites sampled in this region is very low (e.g. 1-4 per site) and indeed does not provide a statistically reliable dataset. It is thus very difficult to make meaningful interpretations on pottery from these sites and a caveat should thus be applied.

Page 16, lines 17-19. 'Several vessels from Grube-Rosenhof were characterized by long-chain wax esters and/or ranges of long chain even-numbered fatty acids and alcohols indicative of degraded plant waxes'

I would point out that whilst long chain even-numbered fatty acids and alcohols can be indicative of degraded plant waxes, in combination, the presence of long-chain wax esters is more suggestive of the processing of beeswax rather than plants. There does not seem to be any detailed description of the distributions of such compounds or, indeed, a chromatogram, to make an assessment of this and I suggest the authors either remove this or clarify.

Oddly, on page 16, line 46-48, the authors state that 'Resins, tars and pitches as well as beeswaxes were comparatively rare despite the fact that these products are readily identifiable in archaeological pottery'. This comment seems odd as beeswax has not been mentioned at all in the paper up to that point. See my comments above though where they claim wax esters are indicative of plant processing.

Reviewer #3

The paper presents large original data, and therefore larger analysis and interpretation can be acceptably left for future works. I have no major remarks to make, and the manuscript can be published basically as it is. This being said, the question of temporal resolution and reasoning deserves a comment. Starting from the title, the authors discuss about the 'adoption of pottery' –

however, especially the more east you go, the flimsier the temporal framework gets. Despite certain degree of stylistic or technological uniformity, the material may include pottery (of the same type) covering several centuries, even a millennium. Thus, even if the data is in many cases fairly congruous, it's a bold assumption that all analysed pieces would represent the actual 'adoption' phase.

Reviewer comments to Author:

Reviewer: 1

Comments to the Author(s)

Title: is written as an assertion. It should be written better. It is also necessary to specify which region of Europe is being worked on. The work focuses on the different culinary uses of ceramics from different regions of the Baltic. However, it does not conduct an in-depth discussion of the reasons why these different uses exist. The conclusion outlines an idea but does not discuss it in depth.

Keywords: Some of them already appear in the title and others refer to only some of the contexts the article works in (Ertebølle, Narva, Neman). I think they should be changed.

Abstract: it is clear and summarizes the paper very well. However, there is an inconsistency with the chronology mentioned in the text in Materials section. In this last section it is specified that the samples are from the late 6th to the beginning of the 4th millennium cal BC but in abstract it says: mostly dating to the late 6th-5th millennium cal BC. The differences are minimal but should leave no room for doubt.

Introduction: it starts with an introduction to the different positions on the incorporation of ceramics in hunter-gatherer societies. In this regard, there is also literature in both North and South America referring to the use of ceramic containers, fatty acids and hunter-gatherers that the authors could refer to.

The problem of the adoption of ceramics in the different regions of the Baltic is clearly stated. An introduction to the methodology and the sample with which it is intended to address this problem is made.

Materials and method:

First of all, it is not possible to understand why the chronology of Dąbki is unsustainable (page 7 line 17-21).

Secondly, as this paper attempts to discuss the initial use of ceramic technology, an important point is to establish that all samples correspond to similar chronological ranges or that the samples considered correspond to the earliest ceramics. The chronological assignment of the samples is difficult to follow in the text, so it is suggested that a column with the corresponding absolute chronology be incorporated into Table S1.

Although most of the information is published, a minimum specification of the context in which the samples were found would be convenient. All come from excavated sites? There are no surface samples or museum collections? Sometimes the contexts can affect the subsequent chemical analysis of the samples.

The criteria by which 12 samples are selected for TAGs investigation is not specified (page 8 line 46-47).

Finally, all the published samples have the same methodological procedures as those used by the authors for the samples presented here?

Results and discussion:

This section is very well ordered and the conclusions obtained are consistent with the data presented. Although the authors mention the low possibility of recording plant processing, they mention that some samples present plant derived compounds. What are these samples? They are only some of Grube-Rosenhof? How many?

Conclusions:

The authors conclude that the analysis of the content of the ceramics revealed broad trends among the subregions and cultural groupings. They consider that the sub-regional variation observed in the use of ceramics cannot be explained simply because of differences in environmental settings and resource availability. They affirm that the use of hunter-gatherer ceramics has been strongly cultural control. This last statement is not well articulated. Cultural control can mean many things and it would be appropriate for authors to develop this idea further.

Some ethnographic citations are also required when they state: as all documented hunter-gatherers practice some form of culturally specific custom for food preparation and consumption often deploying specific material cultures for defined tasks.

References:

There are references that are not complete in supplementary material (N^o5).

Reviewer: 2

Comments to the Author(s)

This is a comprehensive and highly detailed large-scale study using the technique of organic residue analysis to examine hunter-gatherer pottery use in Neolithic Northern and Eastern Europe.

This paper is a valuable contribution to our understanding of diet and subsistence practices across the Baltic region, demonstrating trends in different resource processing among HG groups. The study has situated the results within the context of the broader region and likely cultural context and I am happy to recommend it for publication subject to a few minor comments as detailed below.

Page 5, line 52. I would comment that this is not the 'largest and most detailed' study of its kind and should not be claimed as such. Previous studies have incorporated larger datasets of potsherds e.g. Evershed et al. 2008 and Whelton et al. 2017 where 2200 and 912 sherds, respectively, were analysed. Mukherjee et al. 2008 also analysed 126 surface residues in a large-scale study.

Page 7 paragraph beginning line 24, I would comment that the number of potsherds analysed from the 18 sites sampled in this region is very low (e.g. 1-4 per site) and indeed does not provide a statistically reliable dataset. It is thus very difficult to make meaningful interpretations on pottery from these sites and a caveat should thus be applied.

Page 16, lines 17-19. 'Several vessels from Grube-Rosenhof were characterized by long-chain wax esters and/or ranges of long chain even-numbered fatty acids and alcohols indicative of degraded plant waxes'

I would point out that whilst long chain even-numbered fatty acids and alcohols can be indicative of degraded plant waxes, in combination, the presence of long-chain wax esters is more suggestive of the processing of beeswax rather than plants. There does not seem to be any detailed description of the distributions of such compounds or, indeed, a chromatogram, to make an assessment of this and I suggest the authors either remove this or clarify.

Oddly, on page 16, line 46-48, the authors state that 'Resins, tars and pitches as well as beeswaxes were comparatively rare despite the fact that these products are readily identifiable in archaeological pottery'. This comment seems odd as beeswax has not been mentioned at all in the paper up to that point. See my comments above though where they claim wax esters are indicative of plant processing.

Reviewer: 3

Comments to the Author(s)

The paper presents large original data, and therefore larger analysis and interpretation can be acceptably left for future works. I have no major remarks to make, and the manuscript can be published basically as it is. This being said, the question of temporal resolution and reasoning deserves a comment. Starting from the title, the authors discuss about the 'adoption of pottery' - however, especially the more east you go, the flimsier the temporal framework gets. Despite certain degree of stylistic or technological uniformity, the material may include pottery (of the same type) covering several centuries, even a millennium. Thus, even if the data is in many cases fairly congruous, it's a bold assumption that all analysed pieces would represent the actual 'adoption' phase.

Author's Response to Decision Letter for (RSOS-192016.R0)

See Appendix A.

Decision letter (RSOS-192016.R1)

27-Mar-2020

Dear Dr Courel,

It is a pleasure to accept your manuscript entitled "Organic residue analysis shows sub-regional patterns in the use of pottery by Northern European hunter-gatherers" in its current form for publication in Royal Society Open Science. The comments of the reviewer(s) who reviewed your manuscript are included at the foot of this letter.

Please note that the following email address is not recognised, please reply to this email the updated or alternate email address for:

archloze@inbox.lv

on behalf of Professor Matthew Collins (Associate Editor) and Jon Blundy (Subject Editor)
openscience@royalsociety.org

Associate Editor Comments to Author (Professor Matthew Collins):

Thank you for your changes, which have make the text more comprehensible.

You have addressed the points raised by the referees with clarity.

I have no further recommendations to make
Page 27 line 17 Plantago lanceolat is (I think) Plantago lanceolata

Appendix A

Dear Professor Collins,

We thank the three referees for their reviews and welcome the opportunity to respond to some of the concerns raised. In light of these, we have made some modifications to the manuscript, outlined below, and have provided a detailed point-by-point response.

Should you require any further information, please do not hesitate to contact us.

Yours faithfully,

Dr Blandine Courel

Comments from the Editor

Page 11 line 7 “Each sample was measured in replicate and the standard deviation provided take into account the propagation of uncertainties between the replicate measurement of: (i) the sample, (ii) the methylated standard, and (iii) the C16:0 and C18:0 fatty acids standard measured offline.”

What does “in replicate mean” can you clarify?

Response: The majority of the samples were analysed in duplicate. When necessary, some samples were analysed more than twice. This has been modified in the text.

In the same sentence

“standard deviation provided take into”

reads better as

“standard deviation provided takes into”

Response: In light of the comment, we have amended this in the text.

Reviewer #1

Title: is written as an assertion. It should be written better. It is also necessary to specify which region of Europe is being worked on. The work focuses on the different culinary uses of ceramics from different regions of the Baltic. However, it does not conduct an in-depth discussion of the reasons why these different uses exist. The conclusion outlines an idea but does not discuss it in depth.

Response: We believe that the current title carries more weight and impact than a generic title. Regional variability is also an assertion that we wish to make and one we believe is supported by the data. As hunter-gatherers of the Baltic are the only European hunter-gatherers to use pottery, we hoped that this would be clear, but we agree with the reviewer that more detail is required.

Considering also the comments regarding the term adoption, we have changed the title to:

“Organic residue analysis shows sub-regional patterns in the use of pottery by Northern European hunter-gatherers”

Keywords: Some of them already appear in the title and others refer to only some of the contexts the article works in (Ertebølle, Narva, Neman). I think they should be changed.

Response: In light of the reviewers comments, we have altered the keywords to the following:

cooking pottery; hunter-gatherers; organic residue analysis; circum-Baltic region; Late Mesolithic; Early Neolithic

Abstract: it is clear and summarizes the paper very well. However, there is an inconsistency with the chronology mentioned in the text in the Materials section. In this last section it is specified that the samples are from the late 6th to the beginning of the 4th millennium cal BC but in abstract it says: mostly dating to the late 6th-5th millennium cal BC. The differences are minimal but should leave no room for doubt.

Response: We think that our abstract is not inconsistent with the main text. Indeed, the focus of the paper is primarily the earliest phases per culture as indicated in the abstract (“mostly dating”). However, because a few samples were derived from sites with poor chronological control, especially those from the southeastern Baltic region and adjacent areas, some may indeed be younger as described in the *Material and Methods* part. Precise dating of these assemblages in fact represents an enormous challenge, due to a lack of securely associated and suitable contextual material (herbivore bones etc.) and/or carbonised organic residues (foodcrusts) on pottery, particularly in the southeastern Baltic, and due to the fact that foodcrusts derived from aquatic species will be subject to radiocarbon reservoir effects, often in the order of hundreds of years.

Introduction: it starts with an introduction to the different positions on the incorporation of ceramics in hunter-gatherer societies. In this regard, there is also literature in both North and South America referring to the use of ceramic containers, fatty acids and hunter-gatherers that the authors could refer to.

Response: References have been added in the text regarding the use of pottery in hunter-gatherer communities in the Americas and in Africa: Eerkens et al., 2002; Jordan and Zvelebil, 2009; Taché and Craig, 2015; Dunne et al., 2016.

The problem of the adoption of ceramics in the different regions of the Baltic is clearly stated. An introduction to the methodology and the sample with which it is intended to address this problem is made.

Materials and method:

First of all, it is not possible to understand why the chronology of Dąbki is unsustainable (page 7 line 17-21).

Response: To clarify, we have expanded the sentence to the following:

The proposed appearance of Mesolithic pottery at Dąbki as early as ca. 4850/4700 cal BC [37] is difficult to sustain, given the results of organic residue analysis presented in this paper, which shows that most of the foodcrusts dated at Dąbki were composed of carbon from an aquatic source and therefore subject to a significant reservoir effect; most, if not all of this material may date to the second half of the 5th millennium BC.”

Secondly, as this paper attempts to discuss the initial use of ceramic technology, an important point is to establish that all samples correspond to similar chronological ranges or that the samples considered correspond to the earliest ceramics. The chronological assignment of the samples is difficult to follow in the text, so it is suggested that a column with the corresponding absolute chronology be incorporated into Table S1.

Response: We have added the date ranges for each pottery tradition to Table S1. Many of the assemblages sampled cannot be dated more precisely, and accurate dating of individual assemblages is challenging for the reasons given above (response to comment on abstract), so we would not attempt to indicate date ranges for each site in a separate column. Given the number of sites sampled, our results should include the earliest ceramics in each region, even if we don't know which assemblages are the oldest (which is not important if all the assemblages in one region show the same pattern in pottery function).

Although most of the information is published, a minimum specification of the context in which the samples were found would be convenient. All come from excavated sites? There are no surface samples or museum collections? Sometimes the contexts can affect the subsequent chemical analysis of the samples.

Response: The majority of the samples either derive from museum archives or from recent excavations. The samples from Gamburg Fjord and Hjarnø are the only instances in which we have little stratigraphical control as they were found on the seafloor. That being said, their attribution to the Late Mesolithic Ertebølle culture was based on typology. Materials for which conservation treatment are attested (presence of glue for example) were set aside. For the pottery selected, the most external layer was removed in order to eliminate any contamination from the soil or potentially added during the handling of the object and was used as a contamination control sample. To clarify, we have written the following sentences in the section entitled ‘Lipid extraction’:

Prior to lipid extraction, the external layer of each potsherd was removed in order to reduce the potential for contamination from the burial environment or during post-excavation. Moreover, a method blank and standard were extracted alongside each batch of samples to identify any contaminants introduced during the extraction.

The criteria by which 12 samples are selected for TAGs investigation is not specified (page 8 line 46-47).

Response: The samples were selected based on either of the following criteria:

(1) for some samples, the $\Delta^{13}\text{C}$ ($\text{C}_{18:0}\text{-C}_{16:0}$) values were $<-3.3\text{‰}$ (meeting the established criteria for dairy products as proposed by Craig et al., 2012) and it was our hope to corroborate this potential presence.

(2) for others, the lipid distributions obtained (from the lipid extracts) revealed a significant proportion of *n*-alkanes, alcohols and acids that may indicate the presence of degraded plant products or beeswax. This has been specified in the text as well as in the column “Pretreatment” in Dataset 1.

Finally, all the published samples have the same methodological procedures as those used by the authors for the samples presented here?

Response: The majority of the published data (Robson, 2015, Oras et al., 2017, Papakosta et al., 2019) was obtained using the same acid extraction procedure described in the present paper. However, the data reported in Craig et al. (2007, 2011) were obtained by solvent extraction. The majority of these samples were re-extracted using the same acid extraction protocol described in this study.

Results and discussion:

This section is very well ordered and the conclusions obtained are consistent with the data presented. Although the authors mention the low possibility of recording plant processing, they mention that some samples present plant derived compounds. What are these samples? They are only some of Grube-Rosenhof? How many?

Response: Initially, we only reported plant lipids for samples from 14 sites (Asaviec 4, Biarešča 4, Daktariškė 5, Glūkas 3, Gribaša 4, Grube-Rosenhof LA 58, Iča, Kamen' 6, Kretuonas 1, Osa, Rusakova, Varėnė 10, Zacennie, Zvidze). That said, whilst we searched for them in the majority of the samples they were simply absent. However, in light of the above comment, we felt it necessary to include the plant lipids that were present in the samples from the site of Dąbki. These data are reported in Dataset 1.

Over 40 samples from Grube-Rosenhof and 35 samples from Dąbki had plant lipids or traces thereof, including triterpenes, amyirin and their derivatives, lupane derivatives, *n*-alkanes, etc. (cf. Dataset 1). Moreover, in the pottery from the other sampled cultural groups (e.g. Narva and Neman), plant lipids, including alkanes, were present though infrequent (see Dataset 1).

Unfortunately, previous studies had not always systematically recorded and/or published the presence of plant lipids meaning that our understanding of plant processing in hunter-gatherer pottery from this area is rather limited. Overall, despite the presence of plant lipids in many of the samples, their presence cannot be used to quantitatively evaluate the scale that plants had been cooked or processed in pottery. In addition, the notable lack of plant lipids does not necessarily mean that they had not been processed in the vessels since plants, in general, have lower quantities of lipids. In order to evaluate the importance of plant processing in hunter-gatherer pottery, more research is required using additional analytical methods (analysis of

macro-remains for example), which we are currently undertaking as a part of the wider INDUCE project.

Conclusions:

The authors conclude that the analysis of the content of the ceramics revealed broad trends among the subregions and cultural groupings. They consider that the sub-regional variation observed in the use of ceramics cannot be explained simply because of differences in environmental settings and resource availability. They affirm that the use of hunter-gatherer ceramics has been strongly cultural control. This last statement is not well articulated. Cultural control can mean many things and it would be appropriate for authors to develop this idea further.

Response: We feel as though we do expand explicitly on this point in the following two paragraphs by introducing the notion of ‘cuisine’.

For example, by the sentence “*So whilst there is scant evidence that the environment or food procurement strategies changed with the advent of pottery, culinary ideas for combining and cooking foodstuffs in ceramic vessels were undoubtedly mutable with adoption motivated by prior beliefs, for example, concerning cooking performance and efficacy or equally notions of novelty and prestige.*”

Some ethnographic citations are also required when they state: as all documented hunter-gatherers practice some form of culturally specific custom for food preparation and consumption often deploying specific material cultures for defined tasks.

Response: We agree that more ethnographic studies on hunter-gatherer cooking practices are required and we have added three references (Kelly, 2013; Dunne et al., 2018; Anderson 2019).

Response:

References:

There are references that are not complete in supplementary material (N°5).

Response: We have corrected this by adding the 7 missing references to the SI.

Reviewer #2

This is a comprehensive and highly detailed large-scale study using the technique of organic residue analysis to examine hunter-gatherer pottery use in Neolithic Northern and Eastern Europe.

This paper is a valuable contribution to our understanding of diet and subsistence practices across the Baltic region, demonstrating trends in different resource processing among HG groups. The study has situated the results within the context of the broader region and likely cultural context and I am happy to recommend it for publication subject to a few minor comments as detailed below.

Page 5, line 52. I would comment that this is not the 'largest and most detailed' study of its kind and should not be claimed as such. Previous studies have incorporated larger datasets of potsherds e.g. Evershed et al. 2008 and Whelton et al. 2017 where 2200 and 912 sherds, respectively, were analysed. Mukherjee et al. 2008 also analysed 126 surface residues in a large-scale study.

Response: We agree that other publications have analysed larger numbers of sherds but the relative preservation of lipids was much lower than our study, resulting in less data generated. Nevertheless we have changed the text to report this is the largest study of Baltic pottery.

Page 7 paragraph beginning line 24, I would comment that the number of potsherds analysed from the 18 sites sampled in this region is very low (e.g. 1-4 per site) and indeed does not provide a statistically reliable dataset. It is thus very difficult to make meaningful interpretations on pottery from these sites and a caveat should thus be applied.

Response: In order to highlight that the data should be interpreted cautiously given the small number of vessels analysed, we have made changes in the 'Results and discussion' part:

“Although the results need to be cautiously interpreted due to the small number of vessels per site that have been investigated, our residue data provides a first glimpse into the cuisine and consumption practices of the region.”

Page 16, lines 17-19. ‘Several vessels from Grube-Rosenhof were characterized by long-chain wax esters and/or ranges of long chain even-numbered fatty acids and alcohols indicative of degraded plant waxes’

I would point out that whilst long chain even-numbered fatty acids and alcohols can be indicative of degraded plant waxes, in combination, the presence of long-chain wax esters is more suggestive of the processing of beeswax rather than plants. There does not seem to be any detailed description of the distributions of such compounds or, indeed, a chromatogram, to make an assessment of this and I suggest the authors either remove this or clarify.

Response: We have corrected the paragraph along with its subheading to mention the presence of beeswax that is clearly attested in one sample from Grube-Rosenhof. We have also added a chromatogram in Supplementary Material to illustrate this.

Oddly, on page 16, line 46-48, the authors state that 'Resins, tars and pitches as well as beeswaxes were comparatively rare despite the fact that these products are readily identifiable in archaeological pottery'. This comment seems odd as beeswax has not been mentioned at all in the paper up to that point. See my comments above though where they claim wax esters are indicative of plant processing.

Response: See the response above.

Reviewer #3

The paper presents large original data, and therefore larger analysis and interpretation can be acceptably left for future works. I have no major remarks to make, and the manuscript can be published basically as it is. This being said, the question of temporal resolution and reasoning deserves a comment. Starting from the title, the authors discuss about the ‘adoption of pottery’ – however, especially the more east you go, the flimsier the temporal framework gets. Despite certain degree of stylistic or technological uniformity, the material may include pottery (of the same type) covering several centuries, even a millennium. Thus, even if the data is in many cases fairly congruous, it’s a bold assumption that all analysed pieces would represent the actual ‘adoption’ phase.

Response: We partly agree with the referee concerning the temporal resolution. We attempted to sample pottery dating to the ‘initial adoption phase’ but this ‘phase’, depending on the sub-region, is itself poorly defined and also rather arbitrary in duration i.e. from 10s to 100s of years. We cannot be sure that we have sampled the very first pots that appear in any particular sub-region but, in the sampling section, we have given both date ranges and uncertainties of the pots we analysed in relation to current knowledge regarding arrival times. Regional patterns are observable despite the chronological uncertainty meaning that we can still discuss ‘motivations for the adoption’ of ceramics. However, we have changed the title and sections of the text to reflect that the data is really telling us about regional variability and that ‘motivations’ is inferred.

Finally, more details about the method used for mixing models (FRUITS) have been added in the ‘Materials and Method’ section for more clarity and reproducibility.